# Detection of *ESR1* Mutations in Tissue and Liquid Biopsy with Novel Next-Generation Sequencing and Digital Droplet PCR Assays: Insights from Multi-Center Real Life Data of Almost 6000 Patients

**DOI:** 10.3390/cancers17081266

**Published:** 2025-04-09

**Authors:** Srushti Borkar, Fenja Markus, Agnes Oetting, Stefanie Schmidt, Christine Vössing, David Horst, Markus Möbs, Elena I. Braicu, Frank Griesinger, Katja Horling, Katharina Tiemann, Lukas C. Heukamp, Eva-Maria Willing, Claudia Vollbrecht

**Affiliations:** 1Department of Internal Medicine-Oncology, Carl v. Ossietzky University of Oldenburg, Pius-Hospital, 26121 Oldenburg, Germany; srushti.shirish.borkar@uni-oldenburg.de (S.B.); frank.griesinger@uni-oldenburg.de (F.G.); 2Pius-Hospital, Klinik für Hämatologie und Onkologie, Universitätsklinik Innere Medizin Onkologie, Universitätsmedizin Oldenburg, 26121 Oldenburg, Germany; 3Institut für Hämatopathologie Hamburg, 22547 Hamburg, Germany; markus@hp-hamburg.de (F.M.); oetting@hp-hamburg.de (A.O.); schmidt@hp-hamburg.de (S.S.); voessing@hp-hamburg.de (C.V.); horling@pathologie-hh.de (K.H.); tiemann@hp-hamburg.de (K.T.); willing@hp-hamburg.de (E.-M.W.); 4Lungenkrebsmedizin Oldenburg, 26121 Oldenburg, Germany; 5Institute of Pathology, Charité-Universitätsmedizin Berlin, Corporate Member of Freie Universität Berlin, Humboldt-Universität Zu Berlin and Berlin Institute of Health, Charitéplatz 1, 10117 Berlin, Germany; david.horst@charite.de (D.H.); markus.moebs@charite.de (M.M.); 6German Cancer Consortium (DKTK), Partner Site Berlin and German Cancer Research Center (DKFZ), Im Neuenheimer Feld 280, 69120 Heidelberg, Germany; 7Charité-Universitätsmedizin Berlin, Corporate Member of Freie Universität Berlin, Humboldt-Universität Zu Berlin and Berlin Institute of Health, Klinik für Gynäkologie mit Zentrum für Onkologische Chirurgie und Klinik für Gynäkologie, 13353 Berlin, Germany; elena.braicu@charite.de

**Keywords:** HR+ metastatic breast cancer, *ESR1* mutation, endocrine therapy, liquid biopsy, ctDNA, NGS

## Abstract

Resistance mutations in the *ESR1* gene occur commonly in breast cancer patients previously treated with anti-hormonal therapy. Elacestrant is an FDA and EMA approved oral medication that targets cancer cells in *ESR1* positive tumors, offering patients with these mutations a therapeutic opportunity. It is, therefore, necessary to develop and validate sensitive assays that can detect relevant *ESR1* mutations as early as possible to inform therapeutic decisions for patients eligible for treatment using Elacestrant. We show real-world patient data from two pathology laboratories in Germany to highlight the rarity of *ESR1* mutations, discuss different molecular approaches to detect them, and show a general distribution and pattern of occurrence of these mutations in a cohort of 354 breast cancer patients as detected by our NGS panel. The findings reported in this paper are intended to inform future practices and approaches to molecular diagnostics used in making therapeutic decisions.

## 1. Introduction

Hormone receptor positive tumors expressing the estrogen receptor (ER+) or the progesterone receptor are the predominant forms of breast cancer, accounting for about 80% of all cases [1]. For patients diagnosed with ER+/HER2- breast cancer, targeting estrogen synthesis with anti-endocrine therapy (ET) is the recommended treatment strategy both at initial and metastatic stages. The ET options for ER+ primary breast cancer include aromatase inhibitors that block the conversion of androgen to estrogen, or selective estrogen receptor modulators that antagonize ER activation by reducing co-factor binding [2]. Depending on the molecular profile of the tumor, ETs may also be administered in combination with inhibitors for therapeutic targets, such as cyclin-dependent kinase 4/6 (CDK4/6), mammalian target of rapamycin (mTOR), or phosphatidylinositol-4,5-bisphosphate 3-kinase catalytic subunit alpha (PIK3CA) [3,4,5,6,7].

While ETs are shown to significantly reduce the risk of recurrence from early-stage disease and improve outcomes in patients with advanced disease, more than 30% of ER+ breast tumors ultimately become resistant, leading to recurrence and metastasis [8,9,10]. Notably, mutations in the *estrogen receptor 1* (*ESR1*) gene, particularly those occurring in the ligand binding domain (LBD), have emerged as a mechanism of resistance [11]. Mutations in *ESR1* decrease sensitivity to ET by conferring constitutive activity to the ER, leading to ET resistance and disease progression [12]. In patients treated with ET, *ESR1* mutations are associated with worse progression free survival (PFS) and overall survival, highlighting the clinical relevance of *ESR1* as a biomarker for metastatic breast cancer (mBC) management [13].

Notably, *ESR1* mutations are rarely found in primary tumors (<1%) but are detected in about 25–37% of ER+ mBC cases after ET and may appear at any time under the pressure of treatment [11,12,13,14,15,16,17]. A longitudinal analysis of breast cancer patients showed distinct *ESR1* mutated clones that exhibited divergent behavior over time, strongly suggesting clonal heterogeneity [17]. This emphasizes the need for the longitudinal monitoring of *ESR1* mutations in patients undergoing ET, using methods that can capture tumor heterogeneity. Circulating tumor DNA (ctDNA) from a liquid biopsy (LB) is particularly well suited for this purpose, and several studies have shown that *ESR1* mutations could be detected from a blood-based LB months before clinical disease progression, and the polyclonal nature of breast cancer metastases can be better ascertained with a LB [13,14,18,19].

As of 2023, the Food and Drug Administration (FDA) and European Medicines Agency (EMA) have approved the first oral selective ER degrader, Elacestrant, for the treatment of patients with ER+/HER2- advanced or mBC with *ESR1* resistance mutations [20]. This approval presents diagnostic challenges, as it has defined mutation testing on LBs rather than tissue-based as the diagnostic standard. In the EMERALD trial, patients treated with Elacestrant showed significant improvement in PFS compared to standard of care therapy and a predictable and manageable safety profile [21]. The approval of Elacestrant as a second- or third-line option for patients progressing on ET necessitates the development of accurate and sensitive assays for *ESR1* resistance testing.

Using next-generation sequencing (NGS) data from a large pan-cancer cohort of almost 6000 patients from two German institutes of pathology, we determined the frequency and localization of *ESR1* mutations across various cancer entities. Based on this insight, we designed, validated, and implemented a highly sensitive hybrid capture-based NGS assay covering breast cancer-related genes, including *ESR1*. We compared the NGS assay performance to a commercial digital droplet PCR (ddPCR) assay using reference material as well as clinical samples. Furthermore, we summarized the results of the first 354 consecutive cases analyzed in routine diagnostics.

## 2. Materials and Methods

### 2.1. Molecular Analyses and Patient Cohort

DNA from formalin-fixed paraffin-embedded tissue from a total of 863 patients between 10 years and 89 years of age, including 375 male and 488 female patients with advanced stage cancers, covering 37 different entities were analyzed in routine diagnostics at the Institute of Pathology, Charité-Universitätsmedizin Berlin, Germany with hybrid-capture NGS using the MH Custom Panel V1 (Agilent Technologies, Santa Clara, CA, USA) covering the whole exonic region of 624 genes (Appendix A). The 863 cases evaluated for this study only include cases where the coverage of target regions was >100× for >96% of targets.

For mutation calling, FastQ files were uploaded to MH Guide (Molecular Health, Heidelberg, Germany) and further bioinformatic analysis, annotation, and reporting was performed as part of the MH Guide online platform. After final revision, the results were incorporated into a molecular pathology report and passed to the Charité Comprehensive Cancer Center for the molecular tumor board.

Additionally, DNA from formalin-fixed paraffin-embedded from another 5106 patients, including 2563 male, 2528 female, and 15 of unknown sex, were analyzed at Hämatopathologie Hamburg (HpH), Hamburg, Germany using hybrid-capture NGS panels (HS2-Lung/Lung Liquid, Agilent Technologies; NOGGO GIS V1, Agilent Technologies, Appendix A). Library preparation was performed as described previously [22]. Mutation calling in the exonic regions of target genes was performed using a custom workflow generated in the CLC Workbench (Qiagen, Hilden, Germany), as previously described [23]. In short, the custom CLC workflow for non-liquid assays contains the following steps: 1. Import of FastQ sequences. 2. Extraction of UMIs, which are contained in the first 3 bp of each read, and annotation of each read with UMI information. 3. Mapping of reads. 4. Building UMI consensus reads. 5. Local re-alignment at positions with previously detected structural variants. 6. Locally re-aligned bam is used as input for the low frequency variant caller. 7. Annotation of resulting variants with amino acid changes, exon numbers, splice side effect, and dbSNP information, including CLINVAR.

The custom workflow used for liquid assays incorporates a pre-processing step using fgbio (v1.5.0) and Picard tools (v2.23.1) in order to be able to build double consensus reads. Since UMIs are incorporated at the beginning of each reads in a pair, fragments originating from the forward and reverse strands of the DNA molecule can be differentiated as described in [24]. The following steps are used to generate DCS reads: 1. Extract UMIs (fgbio ExtractUmiFromBam). 2. Mapping of reads against reference hg19 (bwa mem version 0.7.17-r1188). 3. Annotate duplicate reads with duplex UMI information (Picard Tools UmiAwareMarkDuplicatesWithMateCigar version. 4. Build duplex consensus reads (fgbio GroupReadsByUmi and CallDuplexConsensusReads). 5. Format conversion from Bam to FastQ (Picard Tools SamToFastq).

The resulting FastQs are used as input for the custom workflow in the CLC workbench. The workflow is similar to the previously described workflow excluding the UMI handling [24]. The design of this assay does not require paired analysis with normal DNA for mutation detection.

The samples in this cohort covered a broad range of entities, which were retrospectively evaluated for *ESR1* mutations using previously acquired data from the HS2-Lung,/Lung Liquid and NOGGO GIS V1 NGS panels. Any case with a non-synonymous *ESR1* alteration in the LBD was included as an *ESR1*-positive case.

### 2.2. Validation of a Breast Cancer-Specific Liquid Biopsy Assay

For the validation of the custom HS2-Mamma-LIQ assay (Agilent Technologies), the following commercially available reference material with known mutation status was used: Genome in a Bottle Ashkenazim PGP Son Reference Standard (Horizon Discovery, Cambridge, UK), Seraseq ctDNA Complete Mutation Mix AF 0.1% (Seracare, Milford, CT, USA), Seraseq ctDNA Mutation Mix v4 AF 0.5% (Seracare), Seraseq ctDNA Mutation Mix v4 WT (Seracare), ESR1 Reference Set 1% AF cfDNA (SensID GmbH, Rostock, Germany).

To determine threshold values for the required mean sequencing depth (“mean coverage” of the territory) and mapping rate, 381 patient samples were evaluated.

Intra-assay precision was tested by measuring two replicates in two independent sequencing runs. Inter-assay precision was tested by analyzing one diagnostic sample in three independent runs.

To determine the accuracy of the method, reference material containing 16 variants with 0.5% variant allelic frequency (VAF) was evaluated in a total of five analyses (Appendix A).

Further orthogonal validation by comparing the HS2-Mamma-LIQ assay with ddPCR Droplex *ESR1* Mutation Test v2 (Gencurix Inc., Seoul, Republic of Korea) was performed. The ddPCR assay covers 16 *ESR1* hotspot mutations in codons 380, 463, 534, 536, 537, and 538 of exon 5, 7, and 8 with two oligomixes for QX200 Droplet Digital PCR systems (Bio-Rad, Hercules, CA, USA; Appendix A). Therefore, approximately 25 ng cell-free DNA (cfDNA) of 32 patients previously analyzed with HS2-Mamma-LIQ panel were subjected to droplet generation (20,000 droplets/well) and ddPCR according to the manufacturer’s protocol. Patient data were fully anonymized prior to research use. The amplified samples were read in the HEX and FAM channels using the QX200TM droplet reader (Bio-Rad). Reactions with ≥10,000 droplets were considered valid. Data were analyzed using QuantaSoftTM Software v1.7.4 (Bio-Rad) and expressed as copies per well in ddPCR reactions. Thresholds and cut-off values for quantification were set as per manufacturer’s guidelines.

### 2.3. Implementation of the HS2-Mamma-LIQ Assay in Routine Diagnostic

Liquid biopsies of 354 breast cancer patients, including 351 female, 1 male, and 2 patients of unknown sex, were analyzed at HpH using the LB hybrid-capture HS2-Mamma-LIQ assay (Agilent Technologies).

Therefore, two Cell-Free DNA BCT tubes (Streck, La Vista, NE, USA) per patient were collected when *ESR1* status was of clinical interest for therapeutic stratification and cfDNA from plasma was extracted within 14 days of collection either using (i) Roche cfDNA Sample Preparation Kit (Roche, Basel, Switzerland), according to manufacturer’s instructions, or (ii) Maxwell RSC ccfDNA Plasma kit (Promega, Madison, WI, USA), as per manufacturer’s instructions. All cfDNA extractions performed using the Maxwell RSC ccfDNA Plasma kit were pretreated with Proteinase K (Carl Roth GmbH, Karlsruhe, Germany) and 20% SDS (Fisher Scientific, Waltham, MA, USA), and extracted using the Maxwell CSC 48 instrument (Promega). cfDNA was then used in the HS2-Mamma-LIQ assay, and mutation calling using CLC workbench was performed as detailed in Section 2.1.

## 3. Results

### 3.1. Evaluation of the Pan-Cancer Cohort

In order to develop an *ESR1* resistance mutation assay for use in breast cancer patients, we wanted to understand how frequently *ESR1* mutations occur in a pan-cancer cohort without prior ET. We evaluated NGS results of a pan-cancer cohort of 5969 patients from two major German institutes of pathology (for details see Materials and Methods, Section 2).

The HpH pan-cancer cohort includes 5106 cases comprising a range of entities. It features a larger number of lung cancer cases (*n* = 3439, 67%), due to lung cancer patients commonly receiving referrals for molecular diagnostic testing for targeted therapy options. While we observed 29 lung cases to harbor *ESR1* mutations out of 3439 analyzed cases, this represents a frequency of less than 1% (Figure 1a). The only two entities with a larger proportion of *ESR1* positive cases were breast cancer (15%), followed by endometrial carcinomas (7%). It is important to clarify here, however, that the breast cancer cases shown here include all breast entities and are not limited to ER+/HER2- tumors.

*ESR1* positive cases (*n* = 10) in 863 patients tested at the Institute of Pathology, Charité occurred mainly in breast cancer (9/10) and to a lesser degree, in ovarian cancer patients (1/10) (Figure 1b). The remaining 614 patients accounting for all other tested entities harbored no *ESR1* mutation.

These data show that *ESR1* mutations in the non-ET treated patient group are very rare and located throughout the *ESR1* gene. The exact mutations found in this cohort are listed in Appendix A.

### 3.2. HS2-Mamma-LIQ Panel Design and Validation

Based on these findings, we decided to cover not only the so-called hotspot mutations in *ESR1* but the whole exonic regions of the *ESR1* gene with optimal performance for the LBD with the aim of achieving at least a sensitivity comparable to the clinical trial assay of the EMERALD trial. Briefly, HS2-Mamma-LIQ was designed to detect substitutions and indels in 12 genes implicated in breast cancer genes (Table 1) at a limit of detection (LOD) of 0.5% VAF and for known hotspot mutations described in the EMERALD trial with a LOD of 0.1% VAF (Table 2).

A total of 21 different samples for panel validation were successfully processed and sequenced. A total of 66 analyses were performed based on these sequencing data.

The LOD for the HS2-Mamma-LIQ LB analysis was 0.5% or 0.1% VAF for specified *ESR1* and *PIK3CA* hotspot loci (Table 2). To classify a variant as a true positive, we required at least two duplex consensus sequences (DCSs), meaning that the variant was confirmed for both the forward and reverse strands of the original DNA molecule. Using the binomial distribution, a VAF of 0.5% requires 1000× sequencing depth for a theoretical sensitivity of >95% and a VAF of 0.1% requires 4800× sequencing depth for the same theoretical sensitivity of >95%, assuming the duplication rate for on-target reads is close to 100%.

These theoretical values were tested using the reference material (see Section 2.2). To test the LOD of 0.5% VAF, the consensus reads at the loci with known mutations were locally downsampled to a specific sequencing depth. At a sequencing depth of 1500×, all expected mutations with DCS ≥2 could be detected (sensitivity >95%, Table 3). The detection limit of 0.1% VAF was tested using the reference material in a similar manner, and a detection limit of 0.1% VAF with a required sensitivity of 95% was achieved at 7200× sequencing depth.

Of the 381 diagnostic samples evaluated for the determination of threshold values for the required mean sequencing depth (“mean coverage” of the territory) and mapping rate, 245 samples (64%) showed a sequencing depth of ≥1500×, with >95% positions covered by the assay. For these samples, the 5% quantile was determined for each parameter and set as the quality control (QC) threshold (Table 3).

Duplicate analysis for intra-assay precision reached the required QC thresholds. In both duplicates, all 16 variants from the reference DNA were detected as true positive (DCS ≥2, reproducibility 100%, Table 3). The mean measured VAF was 0.5% in the first replicates with a mean deviation of 0.13% and 0.15%, respectively. The mean measured VAF was 0.55% and 0.56% in the other replicates with a mean deviation of 0.20% and 0.17%, respectively.

Inter-assay performance using three samples met the required QC thresholds, and 16 out of 16 variants (DCS ≥ 2) were detected in each sample (repeatability 100%, Table 3). The mean measured VAF was 0.56%, 0.50%, and 0.55%, with a mean deviation of 0.18%, 0.13%, and 0.20%, respectively. The intra- and inter-assay correlation was found to be ≥96%. The precision is, therefore, within the accepted range of ≥95% (Table 3).

To determine the accuracy of the method, one DNA sample from the reference material was examined in a total of five analyses. The material contained 16 variants (SNVs and deletions) with VAF 0.5%. The expected variants were detected in all analyses. There were two variants whose VAF was consistently overestimated (VAF~1%). These were not considered in the following calculations, as they are presumably present in the reference material with a higher VAF. Of the 70 measured VAFs, 66 (94.3%) were within the 95% confidence interval for the respective sequencing depth achieved. For the remaining four data points, the VAF was underestimated. However, all VAFs were well above 0.1% and would have been reliably detected in a normal routine procedure. The VAFs measured by the method are, therefore, within the expected range of the 95% confidence intervals (Table 4).

Orthogonal validation of the HS2-Mamma-LIQ panel with a ddPCR assay (Droplex *ESR1* Mutation Test v2 (Gencurix Inc.)) for 32 samples showed a 96.8% concordance between the two tests for the hotspot mutations in exons 5 (zinc finger domain), 7, and 8 (LBD). NGS results provide information about the particular amino acid change, VAF of the detected mutations, as well as the presence of co-mutations summarized in (Figure 2a).

Due to the ability of the NGS panel to show exact protein changes, we see in patient 10 that multiple *ESR1* mutations were found across three different exons (exon 5, 7, and 8). The ddPCR, however, only detected mutations in exon 8. This is likely due to low allele frequencies of the other two mutations.

Patients who harbor *ESR1* mutations at a low allele frequency may also make ddPCR analysis challenging and lead to false negative results. NGS analysis with the HS2-Mamma-LIQ panel showed that patient 22 had an *ESR1* p.D538G mutation with an allele frequency of 0.1%. A couple positive droplets were observed in the Gencurix ddPCR assay; however, the number of positive droplets did not meet the criteria for a positive result.

### 3.3. Analysis of 354 Consecutive Clinical Liquid Biopsy Samples Using HS2-Mamma-LIQ Assay

A total of 354 consecutive LB samples from breast cancer patients progressing on ET analyzed with the HS2-Mamma-LIQ NGS assay revealed 154 samples (43%) with an alteration in the *ESR1* LBD, and 145 samples (41%) were found to have at least one *ESR1* hotspot mutation (Figure 3).

Further analysis of the distribution of these mutations found that most mutations are clustered between amino acid positions 536–538, making up about 80% of all alterations found. Interestingly, the NGS approach also allowed the detection of other alterations in the LBD which have been previously unreported and are currently characterized as variants of unknown significance. However, it is notable that these alterations in the LBD were not found in the cohort of approximately 6000 cases in our pan-cancer cohort, suggesting that these may be de novo alterations selected under pressure from the ET and may represent *ESR1* resistance mutations. All reported protein changes in *ESR1* are listed in Appendix A.

More than 30% of the patients analyzed with the HS2-Mamma-LIQ assay showed a co-occurrence of pathogenic mutations in *BRCA1/2*, *ERBB2*, *PIK3CA*, and *TP53* (Figure 4a).

Notably, almost 20% of patients had co-occurring *PIK3CA* mutations (Figure 4a) and almost 17% of cases had more than one *ESR1* alteration (Figure 4b). We also found that over 70% of mutations occurred at a VAF under 1.2%, and 40% of these at a VAF ≤0.5%, including 7% at a VAF ≤0.1% (Figure 4c). This is a significant deviation from post-hoc analyses of the EMERALD trial data, which showed that the median VAF was 1.2% [25].

## 4. Discussion

We evaluated NGS analyses of 5969 pan-cancer cases analyzed at two German institutes of pathology. Our data highlight that *ESR1* mutations are extremely rare in ET-naïve cases, present in <1% of cases in entities that are unlikely to have undergone ET. Meanwhile, the elevated proportion of *ESR1* positive cases in breast cancer cases, and to a lesser extent, in uterine cancer, show that these mutations are almost exclusively a consequence of ET. This finding corroborates other data, which advocate for *ESR1* mutations, particularly those in the *ESR1* LBD, to arise as resistance mechanisms to ET [2,26,27,28]. The *ESR1* positive cases in lung, colon, and cholangiocellular carcinoma, and other cancer entities also refer to all variants found in the *ESR1* LBD, most of which are currently specified as variants of unknown significance. These variants were found with a broad range of VAFs, such that no clear pattern was observed. Several of these *ESR1* mutation-positive non-breast cancer cases may have a high tumor mutational burden or microsatellite instability, which may explain the presence of *ESR1* mutations.

The tendency of *ESR1* resistance mutations to appear at any time under the selective pressure of therapy, and their polyclonal nature necessitates a monitoring approach that is sensitive enough to detect them as early as possible, and in a way that tumor heterogeneity can be captured in the case of multiple metastases. Liquid biopsy fulfills both of these criteria, allowing the detection of ctDNA at early stages of progression, from multiple metastases, using a single sample. Importantly, the approval of Elacestrant, which targets *ESR1* mutations, makes it crucial to develop diagnostic assays that are both sensitive and cost efficient to stratify patients harboring these mutations.

Currently, NGS and ddPCR are the mostly commonly used approaches in molecular diagnostics on LB, owing to high sensitivity and reliable variant detection. In this paper, we have introduced our validated NGS assay, the HS2-Mamma-LIQ, optimized to detect mutations in *ESR1* and 11 additional significant genes. While NGS tends to be more cost intensive than ddPCR, we believe that this approach offers a more encompassing view of the tumor’s molecular profile, allowing the detection of alterations at very low VAFs, as well as co-mutations in other clinically relevant genes, such as *PIK3CA*, *BRCA1/2*, *AKT1*, and *ERBB2*. We successfully applied our HS2-Mamma-LIQ assay in routine diagnostics and found 43% of the 354 mBC cases to be *ESR1* mutation-positive, which is at the higher end of the spectrum described in the literature, which reports a prevalence between 25–37% [13,14,15,17]. The high incidence detected in our patients might result from the high sensitivity of our assay and its ability to detect all alterations in the complete LBD. The studies cited above all used a ddPCR approach, which is favored by many in the field due to its low costs and shorter turnover time. However, commercially available digital PCR assays test a selection of commonly occurring hotspots with a multi-plex assay, which may lead to an oversight of other activating mutations. Moreover, over 70% of the *ESR1* alterations we found using the HS2-Mamma-LIQ assay had a VAF under 1.2%. VAFs under 1.2% were markedly greater than those reported in a post-hoc analysis of the EMERALD trial, wherein the median VAF was 1.2% [25]. This may be explained by differences in assay sensitivity, as well as due to variability in the cohort.

As shown by our analysis of 354 patient cases using the HS2-Mamma-LIQ assay, several previously unreported alterations were found in the *ESR1* LBD, which were not found in the large pan-cancer cohort. This may indicate that these alterations have arisen as resistance mechanisms to ET, and future studies could potentially reveal the functional significance of these alterations. An NGS-based approach addresses all of these potential issues, with the added advantage of providing an estimate of VAF, from which clonal heterogeneity can be inferred. Nevertheless, ddPCR may be advantageous as a cost-effective Fast-Track alternative to NGS, particularly in cases of low patient material availability, or for targeted disease monitoring purposes.

In our design of the HS2-Mamma-LIQ panel, we included further therapeutically relevant co-mutations. Particularly, co-mutations found in genes, such as *AKT1* and *PIK3CA,* may be clinically relevant due to the recent EMA approval of Capivasertib for ER+/HER2- mBC patients having mutations in these genes [29]. Notably, a post hoc subgroup analysis from the EMERALD trial shows that, while a clinically meaningful improvement in PFS was associated with Elacestrant compared to standard of care treatment in patients with *ESR1*-mutated tumors who received prior ET and CDK4/6 inhibitors ≥12 months, a slightly shorter PFS was observed among patients with multiple metastases (>3) or those harboring co-mutations in *PIK3CA* or *TP53* [30]. Therefore, a comprehensive assessment of the mutational profile, including clonal heterogeneity and co-mutations, as provided by the NGS-based HS2-Mamma-LIQ assay, can provide clinicians with prognostic insight.

## 5. Conclusions

*ESR1* mutations are exceedingly uncommon in ET-naïve tumors. For breast cancer patients progressing on ET, robust detection methods are necessary to determine treatment options. Whether ddPCR-based or NGS-based, the sensitivity of the assay is particularly critical for early detection and avoiding false negative reports. Based on our findings, NGS-based diagnostic approaches may provide more comprehensive insights into the tumor profile.

## Figures and Tables

**Figure 1 cancers-17-01266-f001:**
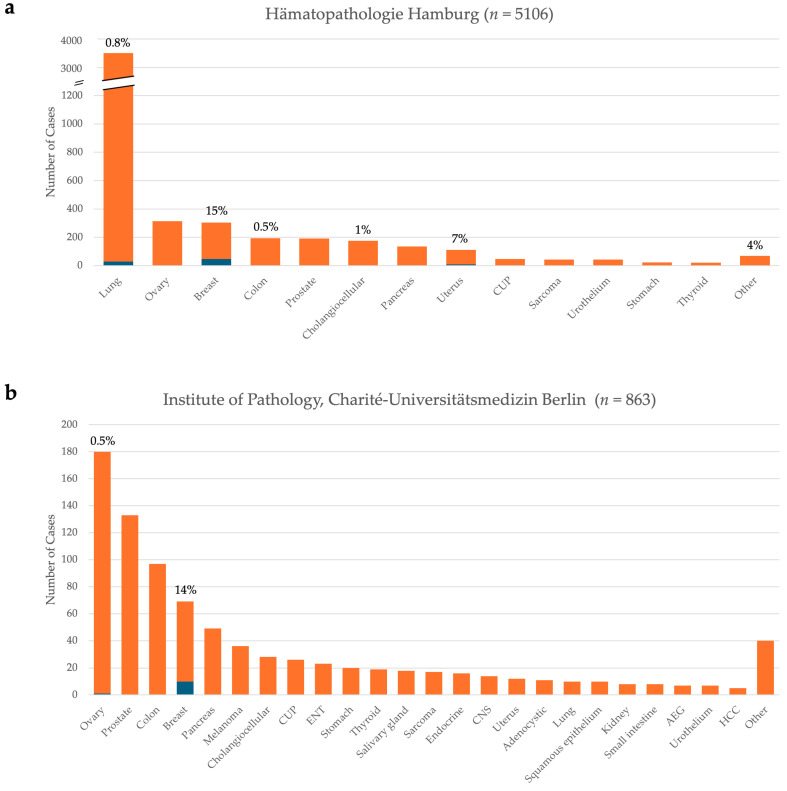
Pan-cancer cohorts from two German institutes of pathologies show that mutations in estrogen receptor gene *ESR1* are exceedingly rare in patients who have not undergone endocrine therapy (ET). (**a**) Retrospective evaluation of *ESR1* mutation status from 5106 cases analyzed at HpH shows that *ESR1* mutations are rarely found in non-breast cancer or non-endometrial cancer entities. Entities with <20 cases are grouped as “Other”, and includes the following: CNS, ENT, duodenum, salivary glands, liver, AEG, adenocystic, melanoma, blood, thymus, vulva, lymph node, parathyroid. (**b**) 863 patients tested at the Institute of Pathology, Charité similarly show that *ESR1* positive cases occur only in breast cancer patients (9/10), and to a lesser degree, in ovarian cancer (1/10). Orange: *ESR1* wildtype cases. Blue: *ESR1* mutated cases. Entities with <5 cases are grouped as “Other”, and include the following: cervix, conjunctival, gallbladder, germ cell, neuroblastoma, squamous cell carcinoma, mesothelioma, thymus, gastrointestinal stromal tumor, and glioma. For entities where the absolute number of *ESR1* positive cases was ≥1, the percentage of positive cases out of the total number of cases for that entity is displayed above the bar. CUP—Cancer of Unknown Primary; ENT—Ear, Nose, Throat; CNS—Central Nervous System; AEG—Adenocarcinomas of the Esophagogastric Junction; HCC—Hepatocellular Carcinoma; HpH—Hämatopathologie Hamburg.

**Figure 2 cancers-17-01266-f002:**
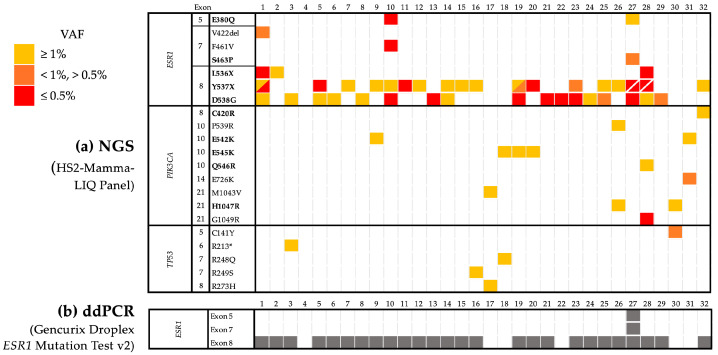
Schematic representation of mutations in the 32 patient cohort used for orthogonal validation. (**a**) Next-generation sequencing (NGS) data show *ESR1* mutated samples with co-occurring *PIK3CA* and *TP53* mutations. Co-occurring mutations are displayed in one box. Mutations that can be detected at 0.1% VAF by the HS2-Mamma-LIQ assay, as shown by the validation data, are marked in bold. Yellow: ≥1% VAF. Orange: <1%, >0.5% VAF. Red: ≤0.5% VAF. (**b**) Digital droplet (ddPCR) data show *ESR1* mutation status for the same 32 cases; Grey: mutation detected.

**Figure 3 cancers-17-01266-f003:**
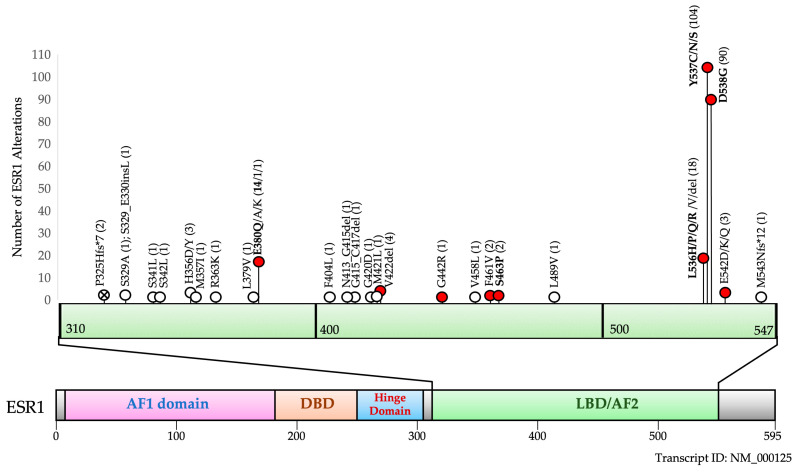
Schematic representation of non-synonymous alterations in the *ESR1* LBD found in 354 consecutive liquid biopsy (LB) analyses of breast cancer patients progressive after ET and/or CDK4/6 inhibitor therapy. A total of 264 non-synonymous alterations in the LBD were found in 354 patients using the HS2-Mamma-LIQ panel. Among commonly occurring hotspot mutations (written in bold) and known activating mutations (red circles), we found several other previously unreported alterations in the LBD. Circles marked with an X refer to alterations that were also found in ET-naïve patients in our pan-cancer cohort. AF1—Activating function 1 domain, DBD—DNA-Binding domain, LBD—Ligand binding domain, AF2—Activating function 2 domain.

**Figure 4 cancers-17-01266-f004:**
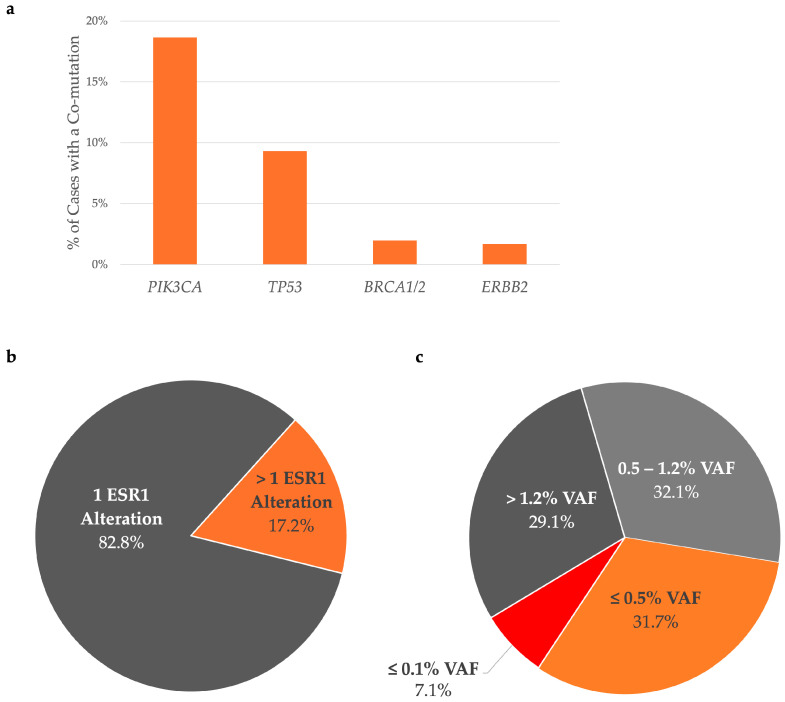
A comprehensive analysis of 354 *ESR1* positive cases using the HS2-Mamma-LIQ assay shows commonly occurring co-mutations. (**a**) Almost 30% of *ESR1* positive patients also show a co-mutation in genes, such as *TP53*, *ERBB2*, *BRCA1*, *BRCA2*, and *PIK3CA*, with almost 20% of cases having a pathogenic *PIK3CA* mutation. (**b**) Of the cases analyzed, almost 17% show a co-occurrence of more than one *ESR1* mutation. (**c**) Over 30% of cases have *ESR1* VAF ≤0.5%, and an additional 7% have VAF ≤0.1%. LBD—ligand binding domain, VAF—variant allele frequency.

**Table 1 cancers-17-01266-t001:** List of genes covered in the HS2-Mamma-LIQ panel.

*AKT1*	*APC*	*BRAF*	*BRCA1*	*BRCA2*
*ERBB2*	*ESR1*	*KRAS*	*NRAS*	*PIK3CA*
*PGR*	*TP53*			

**Table 2 cancers-17-01266-t002:** *ESR1* (NM_000125.4) and *PIK3CA* (NM_006218.4) hotspot mutations covered in the HS2-Mamma-LIQ panel with a limit of detection (LOD) of 0.1% variant allelic frequency (VAF).

*ESR1*	*PIK3CA*
p.E380Q	p.C420R
p.V392I	p.E542K
p.S463P	p.E545X
p.L469X	p.Q546E
p.V534E	p.H1047X
p.P535H	
p.L536X	
p.Y537X	
p.D538G	

**Table 3 cancers-17-01266-t003:** QC thresholds for the HS2-Mamma-LIQ assay.

Million read pairs (mrps)	20
Unique reads	≤20%
Mapping rate	≥91%
Mean coverage (Territory)	≥2800×

**Table 4 cancers-17-01266-t004:** Summary of HS2-Mamma-LIQ panel specifications.

Reproducibility	100%
Repeatability	100%
Sensitivity	≥95%
Limit of Detection (LOD)	0.5%; 0.1% for selected loci (Table 2)
Precision	≥95%
Accuracy of Variant Detection	95%

## Data Availability

The original contributions presented in this study are included in the article and Appendix A. Further inquiries can be directed to the corresponding authors.

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
