# Peer review of "Detection of ESR1 Mutations in Tissue and Liquid Biopsy with Novel Next-Generation Sequencing and Digital Droplet PCR Assays: Insights from Multi-Center Real Life Data of Almost 6000 Patients"

_cancers, 2025, doi:10.3390/cancers17081266_

Round 1
Reviewer 1 Report
Comments and Suggestions for Authors
This is a very good and clinically relevant study.
I have a few suggestions just to improve the clarity for the readers.
- I assume that the 354 cf-DNA samples you sequenced with NGS, you also had FFPE tissue of the same patients sequenced among the large number of pan-cancer samples. If not for all 354, you may have FFPE for majority of them. What was the overlap of mutations among those paired cf- and FFPE samples? In those patients, the PPPE would represent ET-naïve state and cf- may represent post-ET state. This subgroup analysis would strengthen the conclusion you made in this paper. I strongly suggest adding this in the manuscript.
- For sequencing using the MH Custom panel, Please mention what was the median depth of coverage for the 624 target genes so that readers can understand if you had sufficient depth of coverage to identify the low frequency mutations.
- Please clarify the mutation detection parameters using MH guide.
- Same for HS2-Lung/Lung Liquid: mutation detection parameters are not clear. The reference#18 and #19 does not have sufficient detail for mutation detection criteria.
- For the HS2_mamma-Liq sequencing, please clarify that and mention in the main text, if you used only the cf-DNA sample without corresponding normal DNA for mutation detection. In other words, I guess the analysis was not paired. You might have used a common normal pool or have detected the low frequency variants against the reference genome only. Please clarify the custom workflow in CLC workbench in the method section of the main paper.
- For the ddPCR assay, please clarify how many segmentation (e.g 10K, 206K etc) was used per well/sample to detect the variant?
- For the ddPCR assay, how many plex assay was that? Was the multiplex assay designed to capture all the ESR1 variants that you detected by using the HS2-Mamma-Liq NGS assay? If not, mention the one that it had and please clarify exactly which mutations in Exon 5, exon 7 and Exon 8 were supposed to be identified? Figure 2(b) does not allow us to understand.
- Maybe, in Figure 2(a) if you could clarify the Exon#s of the mutations that may help the readers to correlate the ESR1 findings of NGS and ddPCR.
- In figure 3, are all the frequently encountered mutations (>15) in the LBD region covering AA positions 536 – 538 situated in Exon 8? Was the ddPCR designed to identify all those mutations?
- In line 225, 226, you mentioned “A total of 21 different samples for panel validation were successfully processed and sequenced in 31 extractions”. What did you mean? Would you please clarify?
- In line 230: Would you please explain what did you mean by “duplex consensus sequences”? Did you mean that the variant was confirmed in both the forward and reverse sequence reads of the un-broken paired reads?
Reviewer 2 Report
Comments and Suggestions for Authors
This research pertains to the sensitivity of two genetic screening methods, NGS versus standard of care ddPCR assays, to detect somatic ESR1 mutations in breast cancer tissues from patients receiving endocrine therapy. The research results clearly articulated for the most part, and conclusions make a compelling case for the using an NGS method to detect biologically relevant ESR1 mutations in ctDNA samples at levels of sensitivity higher than that that can be achieved by ddPCR. The findings could have implications in monitoring cancer patients for development of resistance to endocrine therapy due to ESR1 mutations.
However, there here are few comments and minor issues that should be addressed to improve the manuscript.
Simple summary section:
Line 28: “data from two pathologies” should be corrected to “data from two pathology laboratories”
Methods section:
Comment: It would be helpful to include a flow diagram as to follow assays performed for each defined group (paraffin-embedded versus liquid biopsy assays) according to various gene panels indicated in Table S1.
Lines 132-134 – Clarify if a separate (additional) ESR1 sequencing assay was performed for all sample groups screened for ESR1 mutations given that all NGS gene panels listed in Table S1 contain ERS1.
Lines 166ff – define “at time of progression under ET” regarding the collecting of cfDNA from plasma from patients.
Results section:
Comment: Though abbreviating hot spot mutations for ESR1 and PIK3CA as shown in Table 2 is acceptable, the mutations listed in supplementary tables S4 S5 and S6 should be provided according to HGSV guidelines (The Human Genome Variation Society, which provides internationally recognized guidelines for describing DNA, RNA, and protein sequence variants). Reporting should include the reference sequence information for each gene sequenced (as shown in Table S3) to facilitate the identification of reported variants in other contexts such as published reports and public databases (such as gnomAD and cBioportal, for cancer genomics, and others)
Comment: While the results of pan-cancer group of nearly 6000 locally derived patients clearly support the rarity of ESR1 mutations in non-endocrine treated cancer patients, further support could be included by surveying genetic variants in the germline, such as from the publicly available Genome Aggregation Database (gnomAD). Population based allele frequencies of ESR1 variants identified in your study should also be included. This would address potential challenges in the interpretation of sequencing results from routine screening for ERS1 mutations purportedly from tumour cells in ctDNA assays and for unique newly identified variants.
Line 234: Table A was not provided.
Re: Table 3: define #mrps
Re: Figure 2a: suggest adding exon number for the location of each ESR1 mutation listed to be able to readily compare with ddPCR results shown in Figure 2b.
Discussion section:
Comment on decision making process when interpretating the clinical relevance of low VAF biologically relevant ESR1 mutations identified during monitoring of patients who have no other indicators of disease progression, given the heterogeneity of breast cancer disease.
